# MP-Mat: A 3D-and-Instance-Aware Human Matting and Editing Framework with Multiplane Representation

**Siyi Jiao**[1*]  **Wenzheng Zeng**[2*†]  **Yerong Li**[1]  **Huayu Zhang**[1]  **Changxin Gao**[1]
**Nong Sang**[1‡]  **Mike Zheng Shou**[2‡]

[1]Key Laboratory of Image Processing and Intelligent Control, School of Artificial
  Intelligence and Automation, Huazhong University of Science and Technology, China
[2]Show Lab, National University of Singapore
{jiaosiyi7, mike.zheng.shou}@gmail.com
wenzhengzeng@u.nus.edu
{yrlee, hyzhang, cgao, nsang}@hust.edu.cn

## Abstract

Human instance matting aims to estimate an alpha matte for each human instance in an image, which is challenging as it easily fails in complex cases requiring disentangling mingled pixels belonging to multiple instances along hairy and thin boundary structures. In this work, we address this by introducing MP-Mat, a novel 3D-and-instance-aware matting framework with multiplane representation, where the multiplane concept is designed from two different perspectives: scene geometry level and instance level. Specifically, we first build feature-level multiplane representations to split the scene into multiple planes based on depth differences. This approach makes the scene representation 3D-aware, and can serve as an effective clue for splitting instances in different 3D positions, thereby improving interpretability and boundary handling ability especially in occlusion areas. Then, we introduce another multiplane representation that splits the scene in an instance-level perspective, and represents each instance with both matte and color. We also treat background as a special instance, which is often overlooked by existing methods. Such an instance-level representation facilitates both foreground and background content awareness, and is useful for other down-stream tasks like image editing. Once built, the representation can be reused to realize controllable instance-level image editing with high efficiency. Extensive experiments validate the clear advantage of MP-Mat in matting task. We also demonstrate its superiority in image editing tasks, an area under-explored by existing matting-focused methods, where our approach under zero-shot inference even outperforms trained specialized image editing techniques by large margins. Code is open-sourced at https://github.com/JiaoSiyi/MPMat.git.

## 1 Introdoction

Human matting is one of the foundation tasks in computer vision that can widely serve for applications such as image editing, image compositing, and film post-production (Zhu et al., 2017; Chen et al., 2018; Sengupta et al., 2020; Lin et al., 2021; 2023). Despite the development of effective algorithms, most methods focus on human matting under single-instance scenarios, which cannot fully align with real-world applications, where multiple instances could exist in a complex scene. In recent years, InstMatte (Sun et al., 2022) formally introduced the multi-instance matting fomulation, which separates the image into a combination of multiple instance layers and background layer:

$$I = \sum_{i=1}^{n} \alpha_i C_i + \left(1 - \sum_{i=1}^{n} \alpha_i\right) B \tag{1}$$

---

[*]Equal contribution, [†]Project lead, [‡]Corresponding author.

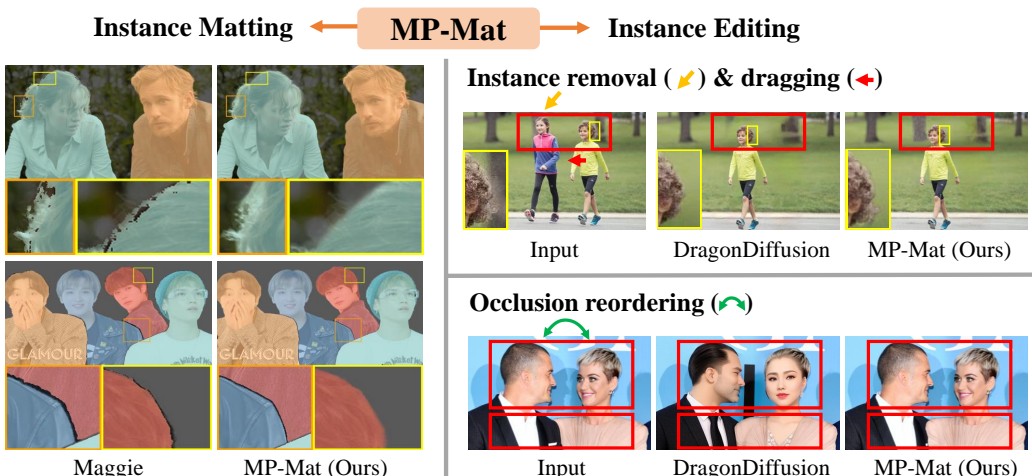

Figure 1: The proposed MP-Mat can perform well in both instance matting and editing tasks, outperforming existing state-of-the-art specialist models designed for individual tasks. Distinguished areas are highlighted with bounding boxes, where MP-Mat preserves finer details and better retains regions that should remain semantically unchanged.

where $\alpha_i \in [0, 1]$ denotes the opacity (alpha matte) of the $i$-th foreground, whose value is the ultimate task goal. The task is actually an ill-defined problem since the foreground color $C_i$, background color B and the alpha value $\alpha_i$ are left unknown. Compared with single-instance assumption, multi-instance matting poses additional challenges. Specifically, the algorithm should be instance-aware (i.e., can localize and distinguish different human instances), and also needs to preserve complex and fine instance edges. Maintaining the integrity of each instance without blurring the edges is particularly challenging in cases where instances are in contact or occluded (Ke et al., 2021).

A core motivation of this work is to establish layered representation to facilitate instance matting in complex scenarios, where we split the scene into different layers based on 2 different perspectives: depth-oriented and instance-oriented. Our design of such layered concept is partially inspired by the idea of Multiplane Image (MPI) (Tucker & Snavely, 2020), a learnable 3D representation that is proposed for novel view synthesis. MPI aims to learn a 3D scene representation with multiple RGBA planes from source image, which can then be used to synthesize different novel views of the scene. The learned multiple RGBA planes can split the scene based on the depth difference. We argue that such a depth-oriented plane-splitting idea could be potentially helpful for matting in multi-instance scenarios. Specifically, different instance may lie in different depth positions, so they can be effectively distinguished by grouping them into different planes that represent a distinct depth level, which could be potentially helpful for handling occlusions under complex scenes. However, we argue that naively using MPI for instance matting is not feasible, as it is build on low pixel-level based on depth differences for only pixel-level novel view synthesis and without instance-awareness.

Based on the aforementioned analysis, we propose MP-Mat, a 3D-and-instance-aware matting framework that is built on meticulously designed multiplane representations. Specifically, our formulation mainly consists of 2 parts: scene geometry-level multiplane representation (SG-MP) and instance-level multiplane representation (Inst-MP). The SG-MP is built to split the scene into multiple planes based on the depth differences, making the scene representation 3D-aware, and can serve as an effective clue for splitting instances in different 3D positions. Different from the existing MPI representations, the proposed SG-MP is built on feature-level, rather than the low pixel-level. The benefits lie in 2 aspects: (1) Compared with low-level RGBA, building MP features on high-level deep features can contain more semantic information to better represent the scene geometry as well as fine-grained texture context, which can be useful for the subsequent instance-level analysis based on it; (2) When optimized together with subsequent instance-level Inst-MP for instance-level perception tasks (e.g., instance localization and matting), the SG-MP feature will also receive relevant gradient, making the plane division and scene representation become more instance-aware.

Besides the built SG-MP, we also introduce an instance-level multiplane representation (Inst-MP) that splits the scene in an instance-level perspective, and represent each instance with both matte and color (as shown in Fig. 2). Our formulation has several benefits: (1) Different from most mat-

ting methods that only predict alpha matte, our proposed representation additionally estimates the color of each foreground, enabling better foreground content awareness, resulting in a higher matting accuracy; (2) Besides foreground instances, Inst-MP also explicitly models the background as a special instance. Such formulation is different from existing works that only focus on foreground matte modeling, and enables better handling of the boundary of instance and background, especially when occlusion occurs; (3) The proposed instance-level multiplane representation obeys the integration property for image rendering (i.e., the integral of the representation equals the whole RGB image), and thus can be easily used for downstream tasks like image editing where instance-level manipulation can be directly processed on separate feature planes. Once built, the representation can be reused to realize controllable instance-level image editing with very high efficiency.

We conduct extensive experiments to demonstrate the superiority of the proposed framework. Specifically, for instance matting, our MP-Mat outperforms existing methods by large margins (i.e., at least 2.76 SAD in HIM-100K dataset (Liu et al., 2024) and 4.16 SAD in SMPMat dataset (Jiao et al., 2024)). Besides, we also take a further step to explore the capacity of MP-Mat on image editing task that is actually under-explored by existing matting-focused methods. We found that our method can also perform well on this potential downstream task and even outperforms existing specialized image editing techniques by large margins and with high efficiency, even under zero-shot inference. A qualitative comparison is shown in Fig. 1, where distinguished regions are highlighted with bounding boxes. For both types of tasks, MP-Mat can actually preserve finer details and better retains regions that should remain semantically unchanged. We hope our work can inspire future research in related fields, including but not limited to image matting and image editing.

## 2  RELATED WORK

### 2.1  INSTANCE MATTING

Elder matting works (Xu et al., 2017; Tan et al., 2018; Lu et al., 2019; Zhang et al., 2019; Qiao et al., 2020; Li & Lu, 2020; Sun et al., 2021; Yu et al., 2021b; Ke et al., 2022; Li et al., 2022; Ma et al., 2023) assume a single-instance condition without multi-instance awareness, which remains a gap with many real-world scenarios. Human instance matting is a recently emerged task that differs from the traditional one, as it requires simultaneously localizing multiple instances and distinguishing their mattes in an instance-level manner. Such a task poses more challenges as it easily fails in complex cases requiring disentangling mingled pixels belonging to multiple instances along hairy and thin boundary structures. Mainstream methods (Hu & Clark, 2019; Sun et al., 2022; Huynh et al., 2024) for this task typically rely on instance-level masks (He et al., 2017; Wang et al., 2020) as input and gradually refine them to predict the matte. While these methods provide a certain level of instance-awareness, it is relatively weak as they largely depend on a pre-set instance segmentation module. Although a recent work (Liu et al., 2024) achieves independent instance-aware capability, its performance still falls short of the SoTA methods in the mainstream setting, primarily due to the lack of external guidance. Different from these works, we build layered representations to emphasize 3D and full instance awareness. Specifically, the build SG-MP decomposes the scene based on depth variation, thereby improving interpretability and boundary handling ability especially in occlusion areas. Our Inst-MP, besides representing the matte of foreground instances, also explicitly models the background and color of each instance, resulting a better context awareness for both foreground and background. Inst-MP also processes good properties for downstream image editing tasks that are under-explored by existing matting-focused methods.

### 2.2  INSTANCE CONTROLLABLE IMAGE EDITING

The task aims to impose instance-level editing (e.g., instance removal, dragging) on the image in a harmonious way. Recently, diffusion models (Yildirim et al. (2023); Shi et al. (2024); Ekin et al. (2024); Sheynin et al. (2024); Yang et al. (2024b); Ren et al. (2024)) have brought significant breakthroughs in editing tasks. Despite the effectiveness, they generally need more denoising steps for high-quality generation, which is usually more time costly. Existing methods are also of weak instance-aware capabilities and cannot fully guarantee that theoretically unchanged regions remain unaffected, resulting in poor performance on instance-level editing tasks. In this work, the intrinsic property of our proposed instance-level multiplane representation (Inst-MP) can also enable an easier and more direct way to achieve instance-level editing, where accurate instance-level feature manipulation can be done on separate feature planes without affecting other regions, thereby enhancing the consistency of the edited image. Experiments show that our approach, even under zero-shot

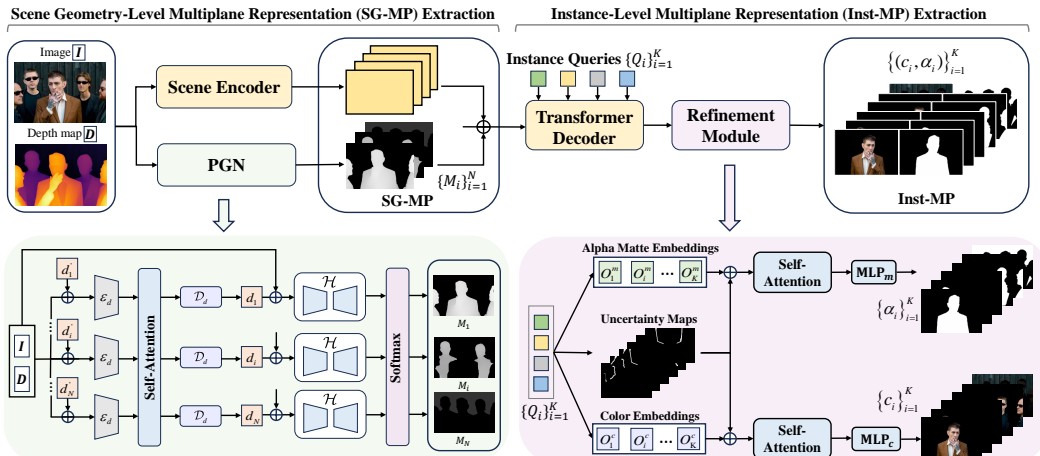

Figure 2: The overall framework of the proposed MP-Mat. ⊕ indicates concatenation operation.

inference, significantly outperforms trained specialized image editing techniques. Another distinguished advantage is that once Inst-MP is built, subsequent editing will take negligible time.

## 2.3 3D SCENE REPRESENTATION

3D scene representation has been widely used for tasks like 3D reconstruction (Mescheder et al., 2019; Wu et al., 2024) and view synthesis (Kong et al., 2024; Luiten et al., 2024). In this work, one of our motivations is to make instance matting become more 3D-aware. Our design is partially inspired by the idea of Multiplane Image (MPI) (Tucker & Snavely, 2020; Zhou et al., 2018), a learnable 3D representation that is proposed for novel view synthesis. MPI aims to learn multiple planes that decompose the scene according to depth variations, which also share similar spirit with some other relevant concepts like layered depth images (LDI) (Shade et al., 1998). We argue that such a depth-oriented plane-splitting idea could be potentially helpful for matting in multi-instance scenarios. However, naively using MPI for instance matting is not feasible, as it is build on low pixel-level based on depth differences for only pixel-level novel view synthesis and without instance awareness. On the contrary, our formulation differs to it within 3 aspects: (1) MPI split the planes based on depth variations, while we design the multiplane representation from 2 different perspectives (scene geometry level and instance level) to be better aware of both depth and instance-level information; (2) MPI is built on low pixel-level, while our scene geometry level representation is built on high-level deep features, which is of better capacity for understanding scene geometry as well as fine-grained texture context. It also enables flexible end-to-end training with the subsequent instance-level task to enrich its instance-awareness ability; (3) MPI is generally used for low pixel-level tasks like novel view synthesis (Tucker & Snavely, 2020) and bokeh rendering (Peng et al., 2022; Rao, 2023) that lack of instance awareness, while our goal is to solve instance-level perception problems, with different formulation from original MPI.

## 3 METHOD

As shown in Fig. 2, our MP-Mat framework mainly consists of 2 parts: scene geometry-level multiplane representation and instance-level multiplane representation. In the following, we will first introduce MP-Mat for instance matting in detail. Then, we will also describe how our method can be applied for instance-controllable image editing tasks.

### 3.1 SCENE GEOMETRY-LEVEL MULTIPLANE REPRESENTATION (SG-MP)

We first build multiplane representations to split the scene into multiple planes based on depth differences. This approach makes the scene representation 3D-aware, and can serve as an effective clue for splitting instances in different 3D positions, thereby alleviating occlusion effects. To be specific, we introduce plane-distinctive masks to represent plane information, which is obtained by a plane generation network. They will be concated with the deep feature encoded from RGBD input, to form the so-called scene geometry-level multiplane representation.

### 3.1.1 PLANE GENERATION NETWORK (PGN)

This network aims to generate the plane-distinctive masks based on the input single view RGBD image, for plane information representation. To be more specific, we first define $N$ planes that aim to split the scene based on distinction in depth and instance-level information, which we represent them using what we call plane-distinctive masks. The PGN will gradually generate $N$ plane-distinctive masks $\{M_i\}_{i=1}^{N}$ based on a set of predefined initial depths $\{d_i'\}_{i=1}^{N}$ (uniformly sampled according to the scene depth map) and the actual RGBD image inputs, where the final refined masks will simultaneously possess scene depth-distinctive and instance-aware characteristics. Note that the input depth map here can be easily estimated using off-the-shelf depth estimators, and the actual cost is comparable to pre-instance mask generation in existing mainstream mask-guided instance matting methods (Sun et al., 2022; Huynh et al., 2024).

As illustrated in Fig. 2, we first use a shared lightweight CNN $\mathcal{E}_d$ to extract a global feature $f_i'$ for plane $i$ at the initial depth $d_i'$:

$$f_i' = \mathcal{E}_d\left(I, D, d_i'\right) \tag{2}$$

Then we apply the self-attention operation to $\{f_i'\}_{i=1}^{N}$ to obtain $\{f_i\}_{i=1}^{N}$:

$$\{f_i\}_{i=1}^{N} = \text{Self-Attention}\left(\{f_i'\}_{i=1}^{N}\right) \tag{3}$$

The intuition here is to adjust the corresponding depth information of each plane at the feature level by exchanging the geometry and appearance information among $\{f_i'\}_{i=1}^{N}$. The adjusted feature $f_i$ is then decoded to the adjusted depth $d_i$ using a shared MLP $\mathcal{D}_d$:

$$d_i = \mathcal{D}_d(f_i) \tag{4}$$

After plane depth adjustment, we then use an interpreter to generate the plane-distinctive masks based on the adjusted plane depths and the original RGBD inputs:

$$\{M_i\}_{i=1}^{N} = \text{Softmax}(\{\mathcal{H}(I, D, d_i)\}_{i=1}^{N}) \tag{5}$$

where $\mathcal{H}$ is a UNet-like architecture, and the spatial resolution of $M_i$ aligns with the extracted deep feature from the scene encoder. The mask delicately allocates every visible pixel from the source viewpoint to each plane. We concat the plane-distinctive masks and the deep scene feature to form the so-called scene geometry-level multiplane representation.

### 3.2 INSTANCE-LEVEL MULTIPLANE REPRESENTATION (INST-MP)

Given the obtained scene geometry-level multiplane representation, we subsequently use it to derive instance-level multiplane representation $\{(c_i, \alpha_i)\}_{i=0}^{S}$, where $i$ serves as the plane index, with $i = 0$ corresponding to the background plane, and $i$ ranging from 1 to $S$ representing instance-level plane for each distinct human instances within the image.

Actually, the goal of instance matting task (i.e., instance-level matte $\{\alpha_i\}_{i=1}^{S}$) is a sub-set of our instance-level multiplane representation. Our formulation has several benefits: (1) Different from most matting methods that only predict alpha matte, our proposed representation additionally estimates the color of each foreground, enabling better foreground content awareness; (2) Different from existing works that only focus on foreground matte modeling, our representation also explicitly model the background by regarding it as a special foreground instance. Such formulation enables better handling of the boundary of instance and background, especially when occlusion occurs; (3) The proposed instance-level multiplane representation obeys the integration property for image rendering: $I = \sum_{i=0}^{S} c_i \alpha_i$. Based on this property, we can reuse the built representation to realize controllable instance-level image editing with very high efficiency (detailed in Section 4). Next, we will introduce how to obtain the instance-level multiplane representation in detail.

### 3.2.1 INSTANCE QUERY

Here we use learnable queries $\{Q_i\}_{i=1}^{K}$ ($K > S + 1$) to capture instance-level features, where $S$ denotes the actual instance amount within image and we also regard background as a special

instance, resulting in the total number as $S+1$. The queries will collect the corresponding instance-level features from the obtained scene geometry-level multiplane representation, through a standard multi-layer transformer decoder Zou et al. (2023) that consists of multi-head self-attention and cross-attention with fully-connected feed-forward networks (FFNs). Then, we use different prediction heads (composed of MLP) to map the query feature $Q_i$ into distinct embeddings: opacity embeddings $O_i^m \in \mathbb{R}^C$ that gauge transparency, and color embeddings $O_i^c \in \mathbb{R}^C$ that capture hue information, where $C$ is the feature dimension. We also derive an uncertainty map $O_i^u \in \mathbb{R}^{C \times H \times W}$ ($H \times W$ corresponds to the spatial resolution of input image) to estimate the uncertainty of the predictions, which will be used for subsequent feature refinement.

### 3.2.2 REFINEMENT MODULE

We design a refinement module that utilizes the uncertainty map $\{O_i^u\}_{i=1}^K$ to enhance $\{O_i^c\}_{i=1}^K$ and $\{O_i^m\}_{i=1}^K$, and finally output the instance-level multiplane representation. Specifically, given the acquired uncertainty map, the top T% (i.e., 10% in our implementation) of pixels with the highest uncertainty values are selected, yielding filtered guidance masks $\{R_i\}_{i=1}^K$ for refinement. The mask $R_i$ is subsequently concatenated with $O_i^m$ and $O_i^c$ to indicate the region of high uncertainty for each instance-level representation. Then, self-attention is adopted among different instance-level features to refine (reconsider) the representations of each instance:

$$\{O_i^m\}_{i=1}^K = \text{Self-Attetion}(\{O_i^m, R_i\}_{i=1}^K)$$
$$\{O_i^c\}_{i=1}^K = \text{Self-Attetion}(\{O_i^c, R_i\}_{i=1}^K) \tag{6}$$

Finally, we predicts the alpha matte $\alpha_i$ and the color $c_i$ based on $O_i^m$ and $O_i^c$:

$$\{\alpha_i\}_{i=1}^K = \text{MattePredictor}\,(O_i^m)$$
$$\{c_i\}_{i=1}^K = \text{ColorPredictor}\,(O_i^c) \tag{7}$$

where both MattePredictor and ColorPredictor are two-layer MLPs that decode the alpha/color embeddings to predict the final alpha matte/color.

### 3.2.3 MODEL TRAINING

The whole MP-Mat framework is trained in an end-to-end manner. We employ a multi-task loss,

$$\mathcal{L} = \lambda_1 \mathcal{L}_{Detect} + \lambda_2 \mathcal{L}_{Matting}$$
$$\mathcal{L}_{Matting} = \mathcal{L}_{alpha} + \mathcal{L}_{lap} + \mathcal{L}_{comp} \tag{8}$$

where $\mathcal{L}_{Detect}$ is the loss for instance detection, including localization and classification loss. The detection predictions are derived from instance-level matte results (the bounding box that can tightly cover the pixel with $m > 0$). We then perform bipartite matching (Carion et al., 2020) between the instance-level detection predictions and GTs to obtain the correspondence between predictions and GTs, and then calculate the standard matting loss $\mathcal{L}_{Matting}$, which encompasses alpha loss, pyramid Laplacian loss and composition loss following standard formulation (Sun et al., 2022).

## 4 MP-MAT FOR INSTANCE CONTROLLABLE IMAGE EDITING

Due to the good properties of Inst-MP as mentioned in Sec. 3.2, MP-Mat can achieve instance-level image editing in a direct, accurate, and nearly free manner. The overarching idea is that instance-level editing can be achieved by directly processing on separate instance-level planes within Inst-MP. Once the Inst-MP is built, subsequent image editing will only take negligible time, and the editing process is training-free. Here we will describe how it works for 3 different sub-tasks in detail.

**Instance removal** aims to delete specified foreground instance from the image without affecting its overall harmony. This problem can be treated as reassigning the alpha matte of the removed instance $j$ to its backward instances (including background). For target instance $j$, We first define $\Omega_j$ as the set of pixel (x,y) where $\alpha_j(x,y) > 0$. For every pixel in $\Omega_j$, we assign its alpha to the instance plane $t$ with $\alpha_t(x,y) > 0$ and is closest behind $j$ (can be estimated by the average depth of the plane):

$$\alpha_t(x,y) = \alpha_t(x,y) + \alpha_j(x,y). \tag{9}$$

Then, the edited image can be represented as:

$$I' = \sum_{i=0, i \neq j}^{S} c_i \alpha_i. \tag{10}$$

**Occlusion reordering** aims to switch the occlusion relationship between the source instance and target instance (e.g., changing from instance $p$ occluding $q$ to $q$ occluding $p$). We first consider a simplified case where no additional instances are positioned between $p$ and $q$ (indicated by depth position) for demonstration purposes. In this case, the task can be done by swapping the alpha values of the two instances within their intersection regions:

$$\alpha_q'(x, y) = \alpha_p(x, y), \tag{11}$$

$$\alpha_p'(x, y) = \alpha_q(x, y), \tag{12}$$

where (x,y) refers to the pixel position that satisfy $(c_q(x, y) > 0) \wedge (c_p(x, y) > 0)$. The edited image can be then synthesized with the updated $\alpha_i$:

$$I' = \sum_{i=0, i \neq p, i \neq q}^{S} c_i \alpha_i + c_p \alpha_p' + c_q \alpha_q'. \tag{13}$$

For general cases where other instances may exist between $p$ and $q$, we first sort the plane based on their depth from the closest to the fastest to the camera plane, resulting in a sorted plane index set $\{..., \mathbf{p}, p+1, p+2, ..., q-1, \mathbf{q}, ...\}$. Then, we perform the aforementioned alpha swap between each pair of adjacent planes iteratively until the desired order is achieved (i.e., $\{..., \mathbf{q}, p+1, p+2, ..., q-1, \mathbf{p}, ...\}$). We provide more detailed description and pseudo code in Appendix A.

**Instance dragging** aims to drag a target instance to a new desired position, which can be divided into two categories: drag across images and drag within one image. The drag across image task can be divided into three steps: (1) Feed the reference image $I_{ref}$ into MP-Mat to get its Inst-MP, and separate the plane $t$ $(c_t, \alpha_t)$ that corresponds to the target instance. (2) Crop $(c_t, \alpha_t)$ to form $(c_t', \alpha_t')$ by extracting the rectangular region that tightly covers the pixels with positive alpha value. (3) Set up a new plane $(c_{new}, \alpha_{new})$ with zero initialization (i.e., all pixel values are 0), and add the cropped $(c_t', \alpha_t')$ to it at the desired position on the target image, resulting in $(c_{new}', \alpha_{new}')$. Note that additional transformations, such as rescaling and rotation, can also be applied to $(c_t', \alpha_t')$ before adding. (4) Add the resulting new plane to the target image $I$:

$$I' = c_{new}' \alpha_{new}' + (1 - \alpha_{new}')I. \tag{14}$$

For dragging within one image, it can be regarded as a combination of instance removal and dragging across images when the target image remains the same as source.

## 5 EXPERIMENTS

### 5.1 TASK, DATASET, AND METRICS

**Instance matting.** We first validate MP-Mat on the multi-instance matting task, where we conduct experiments on 2 datasets: the real image subset of the HIM-100k dataset (Liu et al., 2024), and the SMPMat dataset, a recent synthetic matting dataset. We evaluate different models using two major quantitative metrics: sum of absolute differences (SAD) and mean square error (MSE, we report the $10^2$ scaled value). Lower values for these metrics indicate better alpha matte results. We also report other widely used metrics in Appendix G.

**Instance editing.** This is one of the potential downstream applications for matting, but existing matting-focused research has not explored their actual usability and performance on related tasks. To fill this gap, we conduct extensive experiments on 3 sub-tasks with compelling application value: (1) Instance removal that aims to delete specified foreground instances from the image without affecting its overall harmony; (2) Occlusion reordering that aims to modify the occlusion relationships among foreground instance in a controlled and harmonious manner; (3) Instance dragging that aims to move any instance to the desired position harmoniously. We conduct experiments on the GQA-inpaint dataset (Yildirim et al., 2023) for instance removal task and use Mean L1 Loss, Mean L2

Table 1: Quantitative comparison of instance matting.

| Method | Dataset | | | |
| --- | --- | --- | --- | --- |
| | HIM-100k | | SMPMat | |
| | SAD | MSE | SAD | MSE |
| *Instance-agnostic* | | | | |
| FBA(+Mask RCNN) | 38.25 | 0.95 | 42.36 | 1.12 |
| FBA(+SOLO) | 38.18 | 0.94 | 41.23 | 0.96 |
| FBA(+EVA) | 37.76 | 0.91 | 39.82 | 0.95 |
| MG(+Mask RCNN) | 40.51 | 0.97 | 41.58 | 1.06 |
| MG(+SOLO) | 39.26 | 0.95 | 40.79 | 0.95 |
| MG(+EVA) | 38.19 | 0.94 | 39.73 | 0.95 |
| *Instance-aware* | | | | |
| InstMatte | 37.34 | 0.93 | 40.19 | 0.96 |
| E2E-HIM | 32.22 | 0.84 | 38.55 | 0.93 |
| Maggie | 29.48 | 0.78 | 37.41 | 0.91 |
| MP-Mat (Ours) | **26.75** | **0.49** | **33.25** | **0.81** |

Table 2: Effects of SG-MP and Inst-MP in a global perspective.

| SG-MP | Inst-MP | SAD | MSE |
| --- | --- | --- | --- |
| | | 35.85 | 0.87 |
| | ✓ | 33.78 | 0.86 |
| ✓ | | 29.46 | 0.60 |
| ✓ | ✓ | **26.75** | **0.49** |

Table 3: Components effects within SG-MP extraction.

| Depth | PGN | SAD | MSE |
| --- | --- | --- | --- |
| | | 33.78 | 0.86 |
| ✓ | | 31.82 | 0.74 |
| ✓ | ✓ | **26.75** | **0.49** |

Table 4: Component effect in Inst-MP extraction.

| Background Estimation | Color Estimation | Refinement | SAD | MSE |
| --- | --- | --- | --- | --- |
| | | | 29.46 | 0.60 |
| ✓ | | | 28.53 | 0.55 |
| ✓ | ✓ | | 27.79 | 0.52 |
| ✓ | ✓ | ✓ | **26.75** | **0.49** |

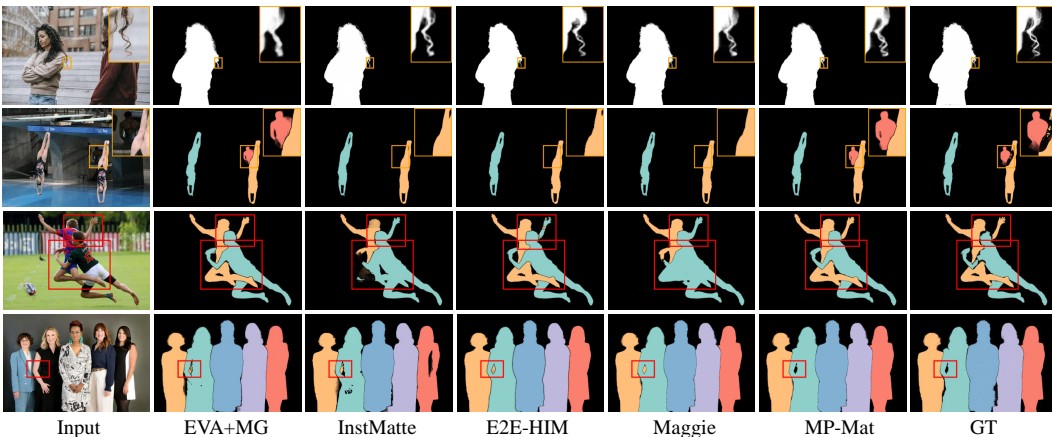

| Input | EVA+MG | InstMatte | E2E-HIM | Maggie | MP-Mat | GT |

Figure 3: Qualitative comparisons. Distinguished areas are highlighted with bounding boxes.

Loss, and PSNR metrics for evaluation, where lower values for Mean L2 Loss and Mean L1 Loss are preferable, while higher PSNR values indicate better quality. For occlusion reordering task, due to the lack of data with ground truth, we construct a synthetic dataset ORHuman that can theoretically ensure the correctness of the derived ground truth (see Appendix C). For instance dragging, we only give qualitative comparisons due to the lack of evaluation benchmarks.

## 5.2 INSTANCE MATTING

**Compared methods.** Our method is compared with the methods designed for this tasks, including InstMatte (Sun et al., 2022), E2E-HIM (Liu et al., 2024) and Maggie (Huynh et al., 2024). We also compared with composite approaches following Liu et al. (2024). Specifically, we tailor the mask-guided matting model MG (Yu et al., 2021a) and FBA (Forte & Pitié, 2020) with off-the-shelf instance segmentation models (Mask-RCNN He et al. (2017), SOLO Wang et al. (2020), and EVA Fang et al. (2023)) to adapt it to the multi-instance matting task.

**Main results.** The main performance comparison on HIM-100K and SMPMat dataset is shown in Tab. 1. It can be observed that the proposed MP-Mat outperforms existing methods by large margins

on both datasets (i.e., at least 2.76 SAD in HIM-100K dataset and 4.16 SAD in SMPMat dataset), demonstrating the superiority of the proposed method.

**Qualitative analysis.** We also give some qualitative results in Fig. 1 and Fig. 3. It can be observed that our method is superior in (1) distinguishing fine-grained boundaries such as hair regions, and (2) with better instance-aware ability, especially under occlusion areas caused by human interactions. More results can be found in Appendix L.

## 5.3 ABLATION STUDIES

Here we conduct ablation studies on the HIM-100K dataset to analyze the effectiveness of the proposed components. We first validate the individual effects of the proposed multiplane representations (i.e., SG-MP and Inst-MP) from a global perspective. Then, we go deeper to analyze the effect of each component within the proposed SG-MP and Inst-MP, demonstrating their effects while also revealing insights for further research. Other studies can be found in Appendix.

**Effects of the proposed multiplane representations.** Here we analyze the individual effect of the proposed scene geometry-level multiplane representation (SG-MP) and instance-level multiplane representation (Inst-MP) from a global perspective. From Tab. 2, it can been seen that: (1) Without SG-MP, the performance drops drastically (SAD from 26.75 to 33.78). This validates the importance of explicit 3D scene representation for multi-instance matting task. Our designed SG-MP makes the scene representation 3D-aware, and can serve as an effective clue for splitting instances in different 3D positions, thereby alleviating occlusion effects; (2) Inst-MP can further boost SAD by a large margin (i.e., 2.71), verifying its effectiveness. Another benefits of such design is for its flexibility and high efficiency on down stream tasks like instance-level image editing, as discussed in Sec. 5.4.

**Component effects within SG-MP extraction.** From Tab. 3, it can be summarized that: (1) Depth information is useful, as it can bring auxiliary 3D information; (2) Only sending depth map as input is not sufficient enough. With our proposed Plane Generation Network (PGN) for multiplane representation at feature level, the performance further boosts by 5.07 in SAD, which is much larger than the performance gain solely from depth input (1.96 in SAD). This essentially demonstrates the effectiveness of our explicit multiplane feature representation extraction design.

**Component effects within Inst-MP extraction.** From Tab. 4, it can be summarized that: (1) Besides focusing on foreground instances, adding explicit background estimation is beneficial for matting accuracy, as it enables better handling of the boundary of instance and background; (2) Besides matte estimation, adding color estimation for each instance can enable better content awareness and thus boost the performance; (3) The proposed uncertainty guided refinement can further facilitate performance, as it can adjust ambiguity at fine-grained boundaries.

## 5.4 INSTANCE EDITING

For all tasks mentioned in Sec. 5.1, we compare our method with SOTA image editing-focused methods (Yildirim et al., 2023; Shi et al., 2024). We finetune the image editing-focused methods on the target dataset for higher performance, while our MP-Mat adopts a zero-shot inference manner.

**Instance removal.** From Tab. 5, we can observe that: (1) MP-Mat significantly outperforms existing editing-based methods and with high efficiency (also refer to Fig. 4 (a) for qualitative comparison). (2) When combined with an off-the-shelf background inpainting model (Yu et al., 2018), the performance of MP-Mat can be further boosted by large margins, we attribute this to the inadequate background modeling capacity of our matting-based methods that is also partially caused by a lack of training data for such themes. (3) The editing time cost within the same image becomes negligible once Inst-MP is constructed in MP-Mat, further verifying the superiority of our design.

**Occlusion reordering.** From Tab. 6, we can also observe the significant superiority of MP-Mat in both effectiveness and efficiency (i.e., more than 10% advantage on Mean L1 Loss, 8.5db on PSNR, and 68% on speed). From Fig. 4 (b), we can observe that when editing target semantics, exiting SOTA methods will also unintentionally alter content that should remain unchanged, such as human faces. In contrast, our method better preserves these elements. The reason is that we conduct editing on individual instance-level planes, where the appearance (color) information remains unchanged, and the only modification is the alpha value (opacity) for new image rendering (users can also inject

Table 5: Quantitative comparison on instance removal. BI refers to an off-the-shelf background inpainting method used to enhance the inpainting of the corresponding region. Bold text indicates the best performance, and underlined text represents the second-best performance.

| Editing Method | Mean L1 Loss (↓) | Mean L2 Loss (↓) | PSNR (↑) | Time per editing (s) | |
|---|---|---|---|---|---|
| Inst-inpaint | 12.69% | 2.58% | 23.09db | 0.1982 | |
| Dragon Diffusion | 12.13% | 2.49% | 22.17db | 0.2139 | |
| Matting Method | Mean L1 Loss (↓) | Mean L1 Loss (↓) | PSNR (↑) | Time per image (s) | Time per editing (s) |
| Ours | 9.37% | 1.96% | 23.58db | **0.1117** | **0.0003** |
| Ours (w/ BI) | **3.73%** | **0.84%** | **25.79db** | 0.1117 | 0.0169 |

Table 6: Quantitative comparison on occlusion reordering.

| Method | Mean L1 Loss (↓) | Mean L2 Loss (↓) | PSNR (↑) | Speed (s) |
|---|---|---|---|---|
| Inst-inpaint | 12.65% | 3.42% | 21.2db | 0.2547 |
| Dragon Diffusion | 13.85% | 3.66% | 19.82db | 0.2849 |
| Ours | **3.26%** | **0.79%** | **28.32db** | **0.1739** |

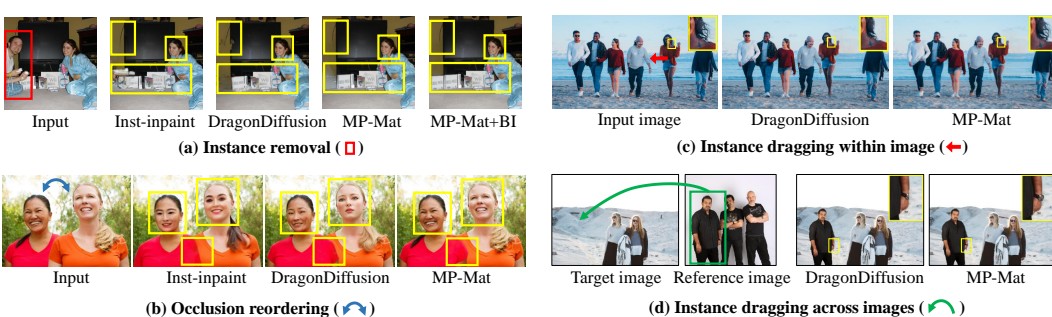

Figure 4: Qualitative comparisons for editing tasks. Yellow boxes highlight the distinguished areas.

some customized transformation like rescaling and rotation, especially for dragging task). This actually provides a better guarantee of not affecting the semantics of areas that should not be altered, further demonstrating the superiority of our approach and highlighting its potential in editing tasks.

**Instance dragging.** Here we only gave qualitative comparison due to the lack of benchmark datasets. As in Fig. 4 (c) and (d), MP-Mat can preserve finer details such as hair and watch after dragging, which highlights another advantage of our approach. Overall, the results highlight the potential of our matting-focused methods for image editing tasks.

## 6 CONCLUSION, LIMITATION, AND BROADER IMPACT

**Conclusion.** In this work, we propose MP-Mat, a 3D-and-instance-aware matting framework that is built on meticulously designed multiplane representations. We design layered representations from two perspectives: the scene geometry-level multiplane representation (SG-MP), which emphasizes scene decomposition based on depth differences, and the instance-level multiplane representation (Inst-MP), which focuses on instance-level modeling. These representations excel in handling occlusion effects and are better aware of both foreground and background content, leading to a significant performance improvement for instance matting. Additionally, our design demonstrates strong potential for instance-level image editing, a relatively underexplored area in existing matting-focused methods. Remarkably, our approach, even under zero-shot inference, outperforms specialized image editing techniques by large margins and with high efficiency.

**Limitation.** Despite the effectiveness, our work mainly focuses on human instances rather than general categories. This is partially due to the lack of data for multi-instance matting for other categories, and we leave this to future works.

**Broader impact.** Besides matting and editing, our layered representation with depth cues may enable extension on other tasks, like 3D photography Shih et al. (2020); Tucker & Snavely (2020), inpainting Shih et al. (2020), distortion correction Wang et al. (2024), etc, where the fine-grained boundary matting and background modeling ability of MP-Mat may bring additional benefits to enhance those relevant tasks.

# 7 ACKNOWLEDGEMENT

Mike Shou is only supported by the Singapore Ministry of Education Academic Research Fund Tier 1 FY2023 Reimagine Research Scheme.

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

APPENDIX

We provide supplementary material for a deeper understanding and more analysis related to the main paper, arranged as follows:

1. More detailed illustration of MP-Mat for occlusion reordering (Appendix A)

2. Implementation details (Appendix B)

3. Details about the constructed ORHuman Dataset (Appendix C)

4. Model's robustness across different depth estimators (Appendix D)

5. More ablation studies (Appendix E)

6. Details on the metric calculation on instance matting (Appendix F)

7. More results under other metrics (Appendix G)

8. Additional evaluation on foreground estimation (Appendix H)

9. Separate evaluation on occlusion and occlusion-free cases (Appendix I)

10. Experiments on instance-agnostic dataset (Appendix J)

11. Generalization capability beyond human category (Appendix K)

12. More qualitative results (Appendix L)

## A  MORE DETAILED ILLUSTRATION OF MP-MAT FOR OCCLUSION REORDERING

In the manuscript, we primarily convey the main editing process of MP-Mat through textual descriptions with part of core equations, due to space limitations. Here, we provide a more detailed illustration of those processes, including additional equations and pseudo code, especially for occlusion reordering.

### A.1  PREREQUISITE

For image $I$ that needs to be edited, we first feed it into our MP-Mat to acquire the Inst-MP representation:

$$R = \{(\alpha_i, c_i)\}_{i=0}^{S} = MPMat(I, D)$$

where $S + 1$ is the prediction instance number (including one for background), and D is the corresponding depth map of $I$. Then, we also estimate the depth of each instance-level plane by averaging the pixels with positive alpha values:

$$\left\{R_i, \bar{d}_i\right\} = \left\{(\alpha_i, c_i), \bar{d}_i\right\}, \bar{d}_i = Avg\left(d\left(\Omega_i\right)\right)$$

where $Avg$ means average operation and $\Omega_i$ is the pixel set containing the positions of the foreground area of plane $i$:

$$\Omega_i = \{(x, y)\}, \alpha_i(x, y) > 0$$

Based on the acquired plane depth $\bar{d}_i$, we sort the Inst-MP $R$ in ascending order of depth, where $R_0$ is the closest instance plane to the camera, and $R_S$ represents the background with its depth considered as infinity. The resulting sorted $R$ will be used for the subsequent editing.

Note that the utilized depth information here can be easily estimated using off-the-shelf depth estimators (e.g., (Yin et al., 2023)), and the actual cost is comparable to pre-instance mask generation in existing mainstream mask-guided instance matting methods (Sun et al., 2022; Huynh et al., 2024).

### A.2  OCCLUSION REORDERING

**Definition:** Switch the occlusion relationship between the source instance and target instance. Here we take: changing from instance $p$ occluding $q$ to $q$ occluding $p$ for example.

Here we focus on the general cases where other instances may exist between $p$ and $q$. The readers can also refer to the manuscript where a simplified case is also provided for easier understanding.

The initial instance index list of $R$ can be represented as: $\{0, 1, 2, ..., \mathbf{p}, p+1, ..., q-1, \mathbf{q}, ..., S\}$

where $R$ has already been sorted based on depth information as illustrated in Sec. A.1. Then, we perform an alpha swap between planes to finally achieve an updated index order:

$$\{0, 1, 2, ..., \mathbf{q}, p+1, ..., q-1, \mathbf{p}, ..., S\}$$

The alpha swap operation can not be done by naively swapping the alpha value between plane $p$ and plane $q$, since the effect from other planes positioned between them will also affect the final rendering. Thus, we decompose this task into iteratively swapping between each pair of adjacent planes until the desired order is achieved. The pseudo-code can be found at Algorithm 1.

---

**Algorithm 1** Algorithm of Occlusion Reordering

# Inputs: Sorted Inst-MP ($R$), instances that require reordering ($p, q$)
# Outputs: new rendered image ($I'$)

**Function** `Swap`$(R, index1, index2)$**:**
    # Identify the intersection area of two instance.
    $\Omega = \{(x, y)| R[index1]['c'](x, y) > 0 \wedge R[index2]['c'](x, y) > 0\}$
    **for** $(x, y) \in \Omega$ **do**
        # Alpha swapping between the intersection area.
        $R[index1]['\alpha'](x, y) \leftrightarrow R[index2]['\alpha'](x, y))$
    **end**
    **return** $R$
**Function** `Rendering`$(R)$**:**
    $I' = 0$
    **for** $index \leftarrow 0$ **to** $len(R) - 1$ **do**
        $I' = I' + R[index]['\alpha'] \cdot R[index]['c']$
    **end**
    **return** $I'$
**Function** `Reordering`$(R, p, q)$**:**
    # Iteratively swap plane $p$ to the target position $q$.
    **for** $index \leftarrow p+1$ **to** $q$ **do**
        $R = $ `Swap`$(R, p, index)$
    **end**
    # Intermediate plane order: $\{0, 1, 2, ..., p+1, ..., q-1, q, p, ..., S\}$

    # Iteratively swap plane $q$ to the target position.
    **for** $index \leftarrow q-1$ **to** $p+1$ **do**
        $R = $ `Swap`$(R, q, index)$
    **end**
    # Final plane order: $\{0, 1, 2, ..., q, p+1, ..., q-1, p, ..., S\}$

    $I' = $ `Rendering`$(R)$

$I' = $ `Reordering`$(R, p, q)$

---

## B IMPLEMENTATION DETAILS

### B.1 GENERAL IMPLEMENTATION DETAILS

For both training and testing, we obtain the input depth map through an off-the-shelf monocular depth estimator (Yin et al., 2023). We train MP-Mat on 4 NVIDIA RTX 2080Ti GPUs with a total batch size of 4 (1 per GPU). The training employs the SGD optimizer with a momentum of 0.9 and a weight decay of 0.0005 for 50,000 iterations. The learning rate is initialized at 0.01 and is adjusted by multiplying with $\left(1 - \frac{iter}{max-iter}\right)^{0.9}$. For the uncertainty map generation, the hyperparameter $K$ is set to 5. In the loss function, the hyperparameters are $\lambda_1 = 1$ and $\lambda_2 = 5$.

### B.2 MORE ILLUSTRATION ON THE LOSS CALCULATION

Here, we provide details regarding the classification loss part of $L_{Detect}$ mentioned in the manuscript. Specifically, this loss is designed to constrain the classification of queries during each network forward pass (i.e., determining whether a query corresponds to a real human instance or a redundant prediction labeled as the "non-object" class). Before calculating the loss, each query is passed through a binary classification head to obtain the predicted 2-class probabilities.

### B.3 PSEUDO LABEL GENERATION IN OCCLUDED AREA

One objective of MP-Mat is to predict foreground color information alongside alpha prediction. For instance editing tasks (i.e., instance removal, occlusion reordering, and instance dragging), awareness of occluded regions is also crucial, as inpainting capabilities are fundamentally required. To this end, we incorporate pseudo-ground-truth RGB content for occluded regions as direct supervision, enabling the model to learn the color information behind occlusions.

We use PowerPaint Zhuang et al. (2024), an inpainting method, to generate such pseudo-labels. Specifically, for each target human instance, we first identify the surrounding instances that may have occlusion relationships with it. Since we do not know which part has occlusion, we treat the closest 4 instances (based on the bounding box center, if they exist) as the surrounding instances. We then merge the ground-truth masks of these surrounding instances into a single mask map that indicates the instances we want to remove (aims to uncover the occluded areas, including the part that may potentially belong to the target instance). We feed such mask along with the original image into the inpainting model, which generates a new image where the surrounding instances are removed, and the occluded areas are inpainted. We then use the acquired new image to generate the new pseudo matting labels (alpha) using an off-the-shelf method Fang et al. (2023). The foreground color label is defined as the RGB pixel value where its corresponding alpha value>0. Note that we only modify the ground truth for the changed areas compared to the original image (i.e., the previously occluded region that is now uncovered), ensuring that the original grouding-truth information remains unchanged. The entire process can be completed automatically without data filtering.

## C    ORHUMAN DATASET

For the occlusion reordering task, due to the lack of data with ground truth, we construct a synthetic dataset ORHuman that can theoretically ensure the correctness of the derived ground truth. The data synthetic pipeline are as follows:

(1) Collect a set of instances $A$ ($A = \{(\alpha_i, c_i) | i \in \{1, 2, \ldots, N\}\}$) along with corresponding high-resolution background image $B$. Here, we collect instances from image matting datasets (Qiao et al., 2020; Xu et al., 2017; Li et al., 2021b), and high-resolution background images without salient objects from the BG-20K dataset (Li et al., 2022).
(2) Randomly generate the overlay order for this set of instances.
(3) Following the synthesis scheme in Sun et al. (2022), overlay the instances onto the background image $B$ in the generated order to obtain the original image $I_o$.
(4) Randomly select instances $p$ and $q$ whose occlusion relationships need to be swapped, and exchange their overlay order during the image synthesis process.
(5) Generate the corresponding instance-swaped image $I_s$ (i.e., the desired ground-truth) according to the new overlay order after the swap.

Based on this schedule, we can obtain diverse data pairs $\{I_o, p, q\} \rightarrow \{I_s\}$ that can be used for both training and evaluation. Some examples are shown in Fig. 5. In total, the constructed ORHuman dataset includes 24,000 training samples, 8,000 validation samples, and 8,000 test samples. Note that in our work, during comparison, our training set is used to fine-tune the existing editing-based methods (Yildirim et al., 2023; Shi et al., 2024) for higher performance, while our MP-Mat has not been trained on it and adopts a zero-shot inference on the evaluation set.

## D    MODEL'S ROBUSTNESS ACROSS DIFFERENT DEPTH ESTIMATORS

Here we further provide experiments to demonstrate the actual robustness of our method and offer an analysis of the reasons behind it. As in Tab. 7, our method is robust to various depth estimators Yin et al. (2023); Yang et al. (2024a); Yin et al. (2019) (spanning various scales and even including

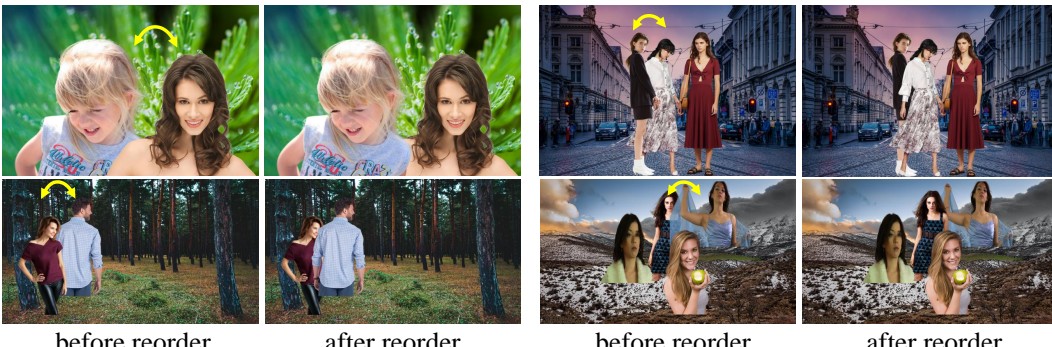

| before reorder | after reorder | before reorder | after reorder |

Figure 5: Examples of the sample pairs within the proposed ORHuman dataset, where we show the pairs of images before and after occlusion reordering (indicated by yellow arrows).

relatively older one published more than 5 years ago, with significantly lower performance than the recent ones). We argue that **such robustness is ensured by the design of MP-Mat itself**, which is in 4 aspects: (1) The plane generation network (PGN) in MP-Mat works in an adaptive manner that gradually refines the plane depth based on not only input depth but also RGB scene information, more evidence can be found in Tab. 2 in the Manuscript, where the performance gain of PGN is much larger than the introduction of depth map itself; (2) we build SG-MP at feature level, which contains not only plane information but also high-level semantic context from the deep feature of RGB image; (3) The precise instance-level perception is ultimately guaranteed by the design of Inst-MP; (4) During optimization, the SG-MP feature will also receive relevant gradient from Inst-MP, making the plane division and scene representation become more instance-aware.

Also, another non-negligible factor is that depth estimation algorithms have already reached a relatively mature stage, even with lightweight ones, and we believe that the future development of depth estimators can further ensure the accuracy of MP-Mat.

| Method | Depth Estimator | Depth Estimation on NYU v2 | | | Instance Matting on HIM-100K | |
|---|---|---|---|---|---|---|
| | | RMSE↓ | Params↓ | Speed↓ | SAD↓ | MSE↓ |
| MPMat | VNL (ICCV'19) | 0.364 | 2.7M | 30ms | 27.24 | 0.51 |
| | Metric3d, used in our work (CVPR'23) | 0.266 | 198M | 183ms | 26.75 | 0.49 |
| | Depth anything v2-ViT-B (Neurips'24) | 0.241 | 335M | 213ms | 26.68 | 0.49 |
| | Mean | - | - | - | 26.88 | 0.50 |
| | Std | - | - | - | 0.22 | 0.01 |

Table 7: Performance under different depth estimators.

# E    MORE ABLATION STUDIES

## E.1    PLANE NUMBER $N$ IN SG-MP

We determine the value based on a parameter search with an initial intuition. Specifically, we believe the number of planes should equal to or larger than the number of people in the scene (including one background). Based on such intuition, we used the average number of instances in our training set (6.95 + 1) as the starting point, and then performed a parameter search. From Tab. 8, it can be observed that: (1) Performance remains relatively stable across different plane numbers; (2) The optimal value is 12, which approximately equals the mean instance number plus one standard deviation (6.95 + 1 + 3.48 = 11.43). Note that we roughly set $N = 10$ in our method; (3) We also report the computational overhead of SG-MP under different plane numbers. It can be observed that the additional cost introduced by more planes is negligible.

## E.2    UNCERTAINTY THRESHOLD $T$

The value is decided by a greedy search. Here we also conduct a experiment to show the performance under different threshold values and provide more insights. The result is listed in Tab. 9. We can observe that: (1) When $T > 0$, the performance improves consistently, which demonstrates the effectiveness of the proposed uncertainty-guided refinement; (2) The performance improvements are relatively stable with different $T$ values. The optimal value is 10%. Here we also provide

| Num_planes | SAD | MSE | Parameters (MB) | Memory usage (GB) |
|---|---|---|---|---|
| 8 | 27.49 | 0.51 | 17.52 | 0.239 |
| 9 | 26.96 | 0.51 | 17.52 | 0.246 |
| 10 | 26.75 | 0.49 | 17.52 | 0.253 |
| 11 | 26.73 | 0.50 | 17.52 | 0.260 |
| 12 | 26.47 | 0.48 | 17.52 | 0.267 |
| 13 | 26.82 | 0.49 | 17.52 | 0.274 |
| 14 | 27.28 | 0.50 | 17.52 | 0.281 |

Table 8: Ablation studies on plane number on the HIM-100K dataset.

some intuitive explanations: We argue that the challenging regions are mainly boundaries among instances, which often exhibit higher uncertainty. However, the ratio of such regions is not very large because boundaries only occupy a small portion of the total pixels of an instance.

| T (%) | SAD | MSE |
|---|---|---|
| 0 | 27.79 | 0.52 |
| 5 | 27.24 | 0.51 |
| 10 | **26.75** | **0.49** |
| 15 | 26.82 | **0.49** |

Table 9: Ablation studies on uncertainty threshold T on HIM-100K dataset.

### E.3 THE SCALING BEHAVIOR WITH THE NUMBER OF INSTANCES

Here we evaluate MP-Mat based on various instance numbers on HIM-100K, where we report MAD (normalized SAD across pixels because SAD is sensitive to pixel numbers) and MSE. From Tab. 10, it can be seen that (1) When instance number>1, a noticeable performance degradation occurs. We argue this is due to the intrinsic challenge of multi-instance scenarios (e.g., the need to distinguish different instances and handle occlusions). (2) The performance is relatively stable when instance number>1, demonstrating the potential working stability of MP-Mat across different scenarios.

| Num_instances | MAD | MSE |
|---|---|---|
| 1 | 3.47 | 0.45 |
| 2 | 5.66 | 0.48 |
| 3 | 5.93 | 0.50 |
| 4 | 5.32 | 0.49 |
| 5 | 5.81 | 0.49 |
| 6 | 5.45 | 0.48 |

Table 10: The scaling behavior with the number of instances on HIM-100K dataset.

### E.4 QUERY NUMBER $K$

From Tab. 11, it can be observed that the matting performance increases as the number of queries grows, although the improvement becomes relatively smaller as the number increases. We set 25 queries in our work.

## F DETAILS ON THE METRIC CALCULATION ON INSTANCE MATTING

Here we will illustrate how we calculate SAD and MSE for instance matting task. Besides matting accuracy, the metric calculation also considers the instance awareness aspect. Specifically, in multi-instance cases, we need to determine the correspondence between each predicted instance-level result and the ground truth instance. Metrics are then computed within each matched pair. This process actually shares similar spirits to the IMQ metric calculation in InstMatte Sun et al. (2022).

More specifically, we first match GT instances with predictions based on IoU (each GT can match at most one prediction, with higher IoU given priority). For unmatched GT instances (indicating missed detection happens), the corresponding matched prediction is set to an all-zero pixel map.

| Num_queries | SAD | MSE |
|---|---|---|
| 15 | 27.94 | 0.52 |
| 20 | 27.55 | 0.51 |
| 25 | 26.75 | 0.49 |
| 30 | 26.36 | 0.49 |
| 40 | 26.08 | 0.48 |

Table 11: Ablation studies on the query number on HIM-100K dataset.

Similarly, for unmatched predictions (indicating false positive), the corresponding matched GT is set to an all-zero pixel map. After the matching process, we can compute SAD or MSE on each GT-prediction pair. In this way, issues such as missed detections and false positives are also reflected in the performance evaluation, in addition to the reflection of instance localization and matting accuracy within matched true positive predictions.

## G    MORE RESULTS UNDER OTHER METRICS

For instance matting task, we provide more metrics here, including Grad, Conn, and IMQ series proposed in InstMatte Sun et al. (2022). From Tab. 12, it can be observed that our method outperforms existing SOTA methods across all metrics consistently, further demonstrating its superiority.

| Method | SAD↓ | MSE↓ | Grad↓ | Conn↓ | IMQmad↑ | IMQmse↑ | IMQgrad↑ | IMQconn↑ |
|---|---|---|---|---|---|---|---|---|
| InstMatte | 37.34 | 0.93 | 18.47 | 37.59 | 56.84 | 73.39 | 56.52 | 58.77 |
| E2E-HIM | 32.22 | 0.84 | 14.51 | 34.82 | 32.97 | 57.43 | 33.24 | 33.95 |
| Maggie | 29.48 | 0.78 | 16.93 | 30.16 | 60.36 | 78.81 | 61.70 | 62.19 |
| MP-Mat (Ours) | 26.75 | 0.49 | 14.26 | 27.83 | 63.75 | 81.49 | 64.26 | 65.44 |

Table 12: Quantitative result under other metrics.

## H    ADDITIONAL EVALUATION ON FOREGROUND ESTIMATION

we conduct experiments on the Composition-1k dataset Xu et al. (2017) whose goal is foreground estimation evaluation. We compare MP-Mat with existing matting methods that have foreground estimation capability (note that all of the existing multi-instance matting methods do not have such capability). The result is shown in Tab. 13, which demonstrates the superiority of our method in foreground estimation.

| Method | SAD↓ | MSE↓ |
|---|---|---|
| SampleNet Tang et al. (2019) | 95.36 | 9.26 |
| FBA Matting Forte & Pitié (2020) | 74.85 | 5.03 |
| Context Aware Matting Hou & Liu (2019) | 61.72 | 3.24 |
| MP-Mat (Ours) | 57.44 | 2.95 |

Table 13: Quantitative results on foreground estimation on the HIM-100K dataset.

## I    SEPARATE EVALUATION ON OCCLUSION AND OCCLUSION-FREE CASES

Here, we propose new metrics to evaluate performance on occluded cases. Specifically, since occlusion is very common and not well-suited for evaluation at the image level, we adopt a more generalized approach by grouping occlusion and occlusion-free cases in a pixel-level manner. Particularly, all predicted pixels are categorized into two groups based on the ground truth (GT): occluded pixels and non-occluded pixels. For a given image, if the GT shows that at least two instances have an alpha value greater than 0 at a specific pixel, that pixel is grouped into the occluded set; otherwise, it will be grouped into the non-occluded set. We then calculate the SAD metric separately within these two sets, resulting in SAD-O for occluded pixels and SAD-NO for non-occluded pixels. The performance is shown in Tab. 14. We observe that our method demonstrates a more obvious advantage in more challenging occluded regions, highlighting the superiority and potential of MP-Mat. (Note that SAD is sensitive to the number of pixels, so the value of SAD-O tends to be lower than that of SAD-NO due to the relatively smaller amount of occluded pixels.)

| Method | SAD-O | SAD-NO | SAD |
|---|---|---|---|
| InstMatte | 18.86 | 18.48 | 37.34 |
| E2E-HIM | 10.40 | 21.82 | 32.22 |
| Maggie | 12.65 | **16.83** | 29.48 |
| MP-Mat | **9.14** | 17.61 | **26.75** |

Table 14: Evaluation on occlusion and occlusion-free cases.

## J EXPERIMENTS ON INSTANCE-AGNOSTIC DATASET

We conduct experiments on the P3M dataset Li et al. (2021a). From Tab. 15, it can be seen that MP-Mat still outperforms existing arts by a noticeable margin. It indeed verifies the superiority of the proposed method.

| Method | Datasets | | | |
|---|---|---|---|---|
| | P3M-500-P | | P3M-500-NP | |
| | SAD | MSE | SAD | MSE |
| InstMatte | 12.49 | 0.0048 | 16.71 | 0.0057 |
| E2E-HIM | 7.96 | 0.0024 | 9.25 | 0.0030 |
| Maggie | 8.94 | 0.0030 | 11.39 | 0.0037 |
| MP-Mat (Ours) | 6.88 | 0.0022 | 8.37 | 0.0028 |

Table 15: Quantitative results on P3M dataset.

## K GENERALIZATION CAPABILITY BEYOND HUMAN CATEGORY

Theoretically, our method can be generalized to instances of other categories (i.e., not limited to human). Some reasons we mainly focus on human instances in our manuscript are: (1) There is a lack of suitable datasets, as no natural multi-instance matting datasets focus on categories beyond humans. (2) Human instance matting itself holds significant application value and can be easily extended to other categories (as long as with sufficient data support). Existing multi-instance matting works also focus on human instances, and we follow this setting by starting with humans.

Here we further conduct experiments to show the generalization ability of MP-Mat beyond human instance. Specifically, we conduct experiment on AM-2K dataset Li et al. (2022), which contains multiple (i.e., 20) instance categories (e.g., cat, dog, camel) but under single-instance scenarios. As in Tab. 16, our method still outperforms other instance matting counterparts by large margins, demonstrating its strong generalization ability across different categories. We believe that the development of high-quality multi-instance datasets will further advance the relevant research.

| Method | AM-2K | |
|---|---|---|
| | SAD↓ | MSE↓ |
| InstMatte | 14.25 | 00053 |
| E2E-HIM | 12.39 | 0.0041 |
| Maggie | 13.83 | 0.0050 |
| MP-Mat (Ours) | 10.57 | 0.0029 |

Table 16: Quantitative results on AM-2k dataset.

## L MORE QUALITATIVE RESULTS

Here we provide more visual comparisons in addition to the examples given by Fig. 1 and Fig. 3 in the manuscript. From rows 1, 2, 4, and 5 in Fig. 6, it can be observed that our MP-Mat preserves finer details (e.g., hair, hands, and other interactive boundaries between instances) better than existing approaches (Yu et al., 2021a; Sun et al., 2022; Liu et al., 2024; Huynh et al., 2024), and in some cases, even surpasses the human-labeled ground truth (Rows 4 and 5). Besides, our method demonstrates strong instance awareness. For example, 3 out of 5 existing methods Sun et al. (2022);

Liu et al. (2024); Huynh et al. (2024) fail to detect instances or assign incorrect pixel associations in rows 3 and 5, while our method robustly distinguishes instances even at a small scale (Row 3) and with challenging boundaries (Row 5). In summary, the proposed MP-Mat excels in instance localization, instance differentiation, and fine detail perception.

During the rebuttal period, we further provide more visualization results, including the visualization of the learned multi-planes and more qualitative examples of more complex hairstyles.

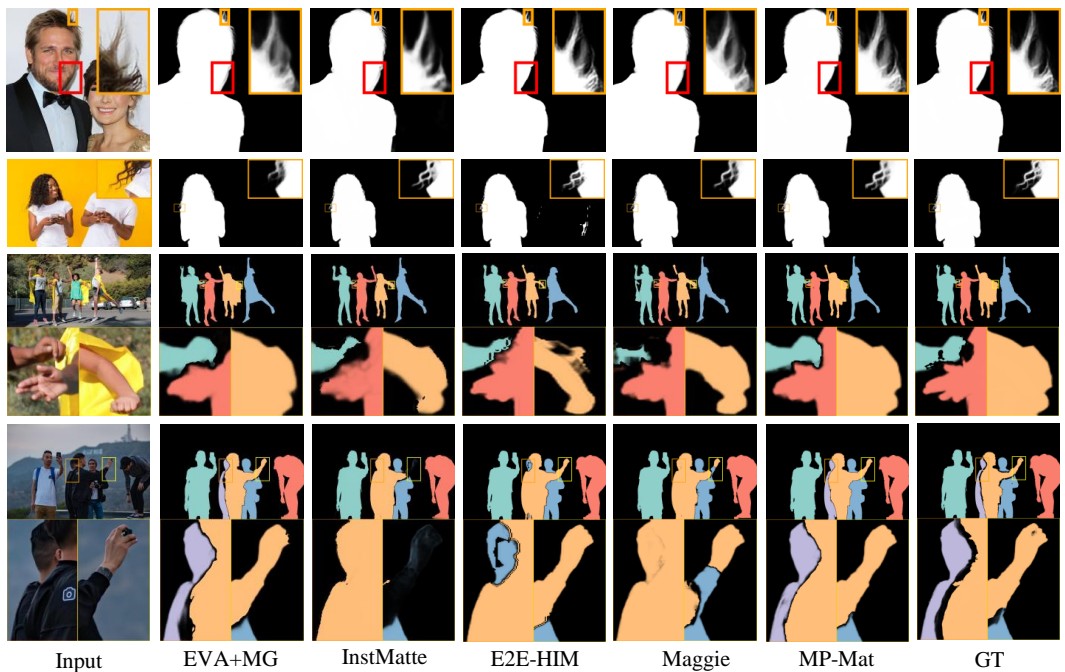

Figure 6: Additional visual comparison on instance matting.

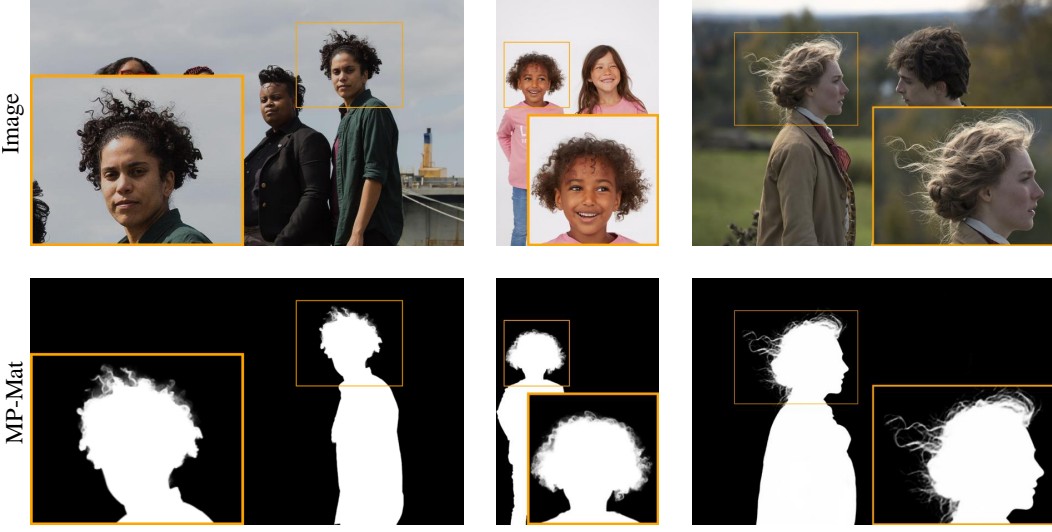

Figure 7: Qualitative examples of more complex hairstyles.

**Qualitative examples of more complex hairstyles.** As in Fig 7, it can be observed that MP-Mat can well handle challenging cases with complex hair hairstyles, further demonstrating its effectiveness.

**Visualization of the learned multi-planes.** From Fig. 8 (a), it can be observed that the learned plane-distinctive masks in SG-MP are reasonable, as they correctly divide the scene based on depth

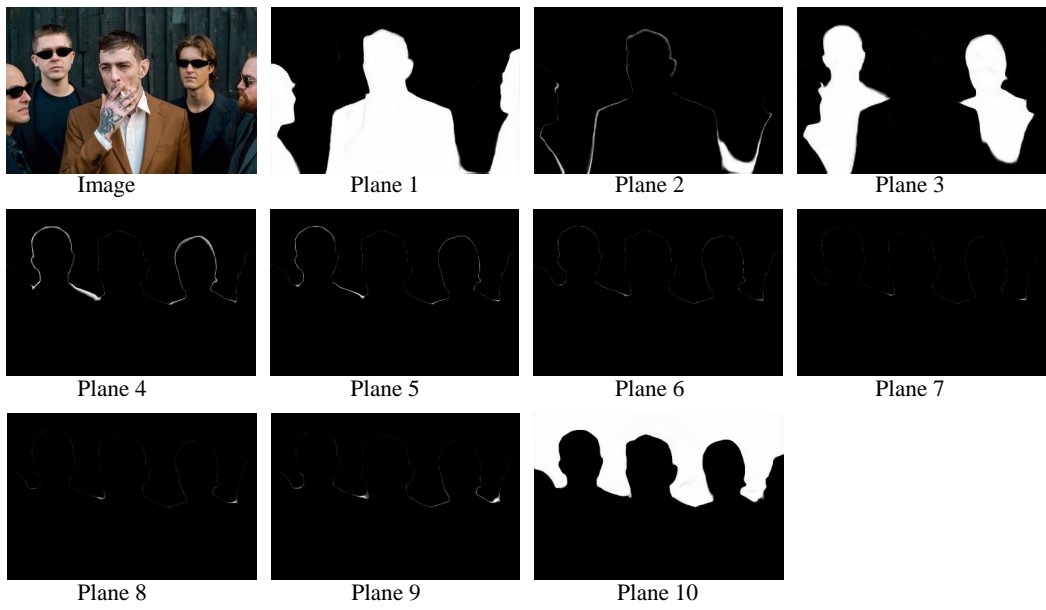

(a) Visualization of the learned plane-distinctive masks in SG-MP.

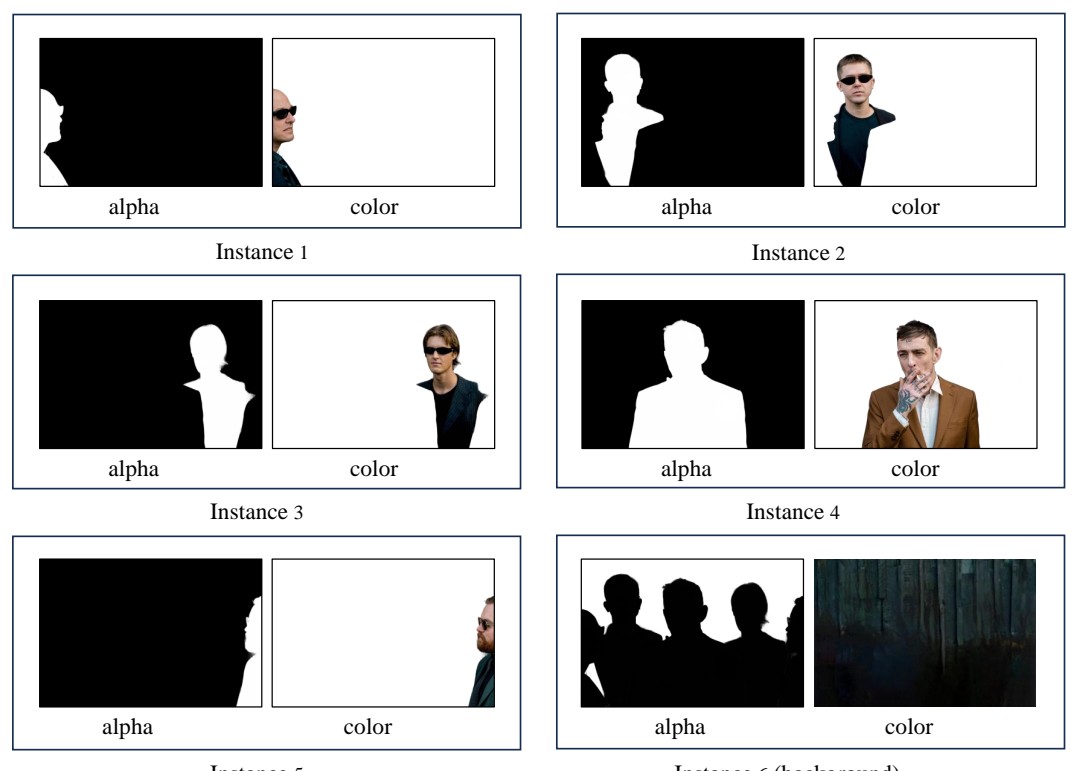

(b) Visualization of the learned Inst-MP representation.

Figure 8: Visualizations of the learned representations in SG-MP and Inst-MP.

variance. Moreover, as shown in Fig. 8 (b), our Inst-MP precisely extracts each instance's alpha values and foreground information (color), providing a solid and efficient foundation for subsequent image editing tasks.

