# OpenReview forum: "MP-Mat: A 3D-and-Instance-Aware Human Matting and Editing Framework with Multiplane Representation"
_ICLR.cc/2025/Conference — ICLR 2025 Poster_

### Official Review · Reviewer_Vmaq · 2024-10-15

**Soundness:** 3
**Presentation:** 2
**Contribution:** 2
**Rating:** 6
**Confidence:** 5

**Summary:**

This paper presents a new instance matting method that enhances 3D scene representation by constructing feature-level multiplane representations. It also introduces multiplane representation to segment the scene at the instance level. Additionally, foreground prediction refinement is incorporated to supervise and assist the network in learning instance-specific features.

**Strengths:**

1. This paper introduces depth information into the instance matting task, which is technically sound in its approach through the use of multilayer representations.
2. The instance multiplane representation offers a new approach distinct from existing methods like E2E-HIM’s instance guidance and Maggie’s IDEmbedding.
3. The method demonstrates strong performance across two datasets, highlighting its effectiveness.

**Weaknesses:**

1. The innovation in foreground prediction has been discussed in other works such as Context Aware Matting, FBAMatting, and SampleNet, making it less novel.  In FBAMatting, the network predicts the alpha matte, foreground, and background simultaneously, which is similar to the design of the two decoders in this paper, but with the additional prediction of the background. This reduces the novelty of the proposed design.
2. The method description is not entirely clear; for instance, the shape of O_i^u seems inconsistent between lines 273 and 279.  It is recommended to clarify the shape of O_i^u and specify whether it involves a k-dimension.
3. The framework appears closely related to E2E-HIM, particularly with the use of bipartite matching during training, but certain aspects, such as the number of queries S, are not clearly explained. It is recommended that the comparison with E2E-HIM be enhanced by clarifying whether there are differences in the bipartite matching during training and in parameters such as the number of queries. Additionally, an ablation study on the hyperparameters would be beneficial.
4. There seem to be issues with the references, with some entries being repeated.

**Questions:**

The paper uses SAD and MSE as evaluation metrics, but it’s important to clarify how these metrics are calculated.  I think the evaluation for multiple instances may differ from traditional single-instance matting. Additionally, given that InstMatt employs the IMQ metric, it would be helpful to explain why this metric was not used in the current evaluation.

---

### Official Review · Reviewer_Yuvj · 2024-10-21

**Soundness:** 3
**Presentation:** 3
**Contribution:** 3
**Rating:** 6
**Confidence:** 3

**Summary:**

This paper presents MP-MAT, a framework using dual multiplane representations (Scene Geometry-level and Instance-level) for human instance matting. The approach achieves state-of-the-art results in both matting (2.76+ SAD improvement) and image editing tasks, with an uncertainty-guided refinement module enhancing performance in ambiguous regions. Notably, MP-MAT's zero-shot performance in editing outperforms specialized methods while maintaining high efficiency.

**Strengths:**

- Introduces a dual multiplane framework that combines geometric depth understanding with instance-level processing, adapting concepts from view synthesis to matting. The treatment of background as a special instance and uncertainty-guided refinement for ambiguous regions represent significant technical advances.
- Demonstrates comprehensive validation through multiple datasets, achieving substantial improvements (2.76+ SAD on HIM-100K) and efficient runtime. The thorough ablation studies and zero-shot performance on editing tasks surpassing specialized methods provide evidence of the method's effectiveness.
- Presents a well-structured technical solution with clear diagrams and effective visualizations.
- The transparent discussion of limitations and systematic presentation of components enhance understanding and reproducibility.

**Weaknesses:**

- The method is tested exclusively on human instances, lacking validation on other object categories or across different datasets.
- The heavy reliance on external depth estimation creates potential error propagation issues. The paper lacks theoretical analysis of when/why the multiplane approach might fail and doesn't explore how the number of planes affects the performance/efficiency tradeoff.
- Key ablation studies regarding plane number sensitivity and computational overhead from dual representation are missing.

**Questions:**

- What drove your choice for the number of planes in SG-MP? Is performance sensitive to this parameter?
- Could you provide the complete architecture of your plane generation network?
- How did you determine the optimal uncertainty threshold T%?
- What are the minimum GPU memory requirements for storing dual representations?
- Have you tested the method on non-human objects?
- Can you provide quantitative results on instance dragging tasks?
- How does performance degrade with imperfect depth maps?
- What's the scaling behavior with the number of instances?

---

### Official Review · Reviewer_ibAt · 2024-11-02

**Soundness:** 3
**Presentation:** 3
**Contribution:** 3
**Rating:** 8
**Confidence:** 4

**Summary:**

This paper introduces a novel approach to instance matting by incorporating depth maps and multiplane image (MPI) representations with a tailored neural architecture. This design can handle occlusions, demonstrated through evaluations on two human instance matting benchmarks. Additionally, the authors evaluate their matting method for image editing and show advantages over diffusion-based methods.

**Strengths:**

- Integrating depth maps and MPI representations into instance matting is novel and useful for handling occlusions. This approach is conceptually sound and well-motivated.
- The method shows high quantitative performance on two benchmarks and also demonstrates effectiveness in image editing tasks, showing the potential application scope of the proposed framework.

**Weaknesses:**

- Only SAD and MSE metrics are provided, whereas other widely used metrics, such as Grad and Conn, are absent.
- Since the paper emphasizes the significance of foreground estimation, additional metrics evaluating foreground estimation would strengthen the evaluation.
- While quantitative results are promising, there are too few visual examples to fully assess the advantages. And most examples involve relatively simple cases (e.g., individuals with simple hairstyles). Including more challenging examples, such as individuals with complex hairstyles, would provide a more comprehensive evaluation.
- The paper's visual results are also relatively small and make it hard to access the matting quality; larger figures are recommended for better clarity.
- Line 519 claims stronger theoretical guarantees for content preservation. However, this seems to primarily benefit from the regression model, while the compared methods utilize diffusion-based generative models. Therefore, I didn't realize there was any theoretical guarantee for this.
- Typos: In Line 389, "Instance-agnostic" should be "Instance-aware."
- Missing references: (1) a highly related thesis on using MPI for image matting: Single portrait image matting and bokeh effect synthesis via multiplane images, Master Thesis, 2023, HKUST. (2) Occasion-aware segmentation.
- There will be some confusion mentioned in the question section.

**Questions:**

- The proposed method may rely heavily on the quality of the depth estimation algorithm. How might the model's performance vary with different monocular depth estimators?
- How to set up the query during inference?
- Additionally, it's unclear how the method does inpainting for tasks like instance removal (e.g., removing a person as shown in Figure 4(a); without the BI module, this proposed method still can in paint). Similarly, it would be beneficial to elaborate on handling occluded person inpainting (e.g., the person in Figure 4(b)).
- Since there is an advantage in dealing with occlusions, separating the evaluation benchmark into two parts (occlusion-based or occlusion-free) and discussing how the model performs on different categories.
- Based on the above, adding the instance-agostic benchmark (such as PPM) would help show the capability to process details. The samples in the instance matting benchmark are too small to see details.

---

### Official Review · Reviewer_dGBZ · 2024-11-03

**Soundness:** 3
**Presentation:** 3
**Contribution:** 3
**Rating:** 6
**Confidence:** 4

**Summary:**

The paper introduces a  3D- and -instance-aware matting framework with multiplane representations. The multiplane concept is designed from two different perspectives: scene geometry level and instance level. The approach builds feature-level multiplane representations which split the scene into multiple planes based on depth differences. This is followed by further splitting of the planes on an instance-level, at which point each instance is associated with an alpha value and a color value.
The detailed splitting into a complex multiplane representation permits other operations, such as instance-level image editing.

**Strengths:**

The multiplane representation is novel, and the two-phase splitting by depth then by instance makes a lot of sense (in terms of different ways that a multi-plane representation could be built).
The paper extends the MPI technique of Tucker and Snavely (2020) by using deep features rather than raw pixel values.
The detailed splitting into a complex multiplane representation permits other operations, such as instance-level image editing. However, the representation presumably is rich enough to enable other image editing pipeline operations, such as creation of stereo pairs (for 3D movies), or doing video editing. This could be mentioned in the paper and the title of the paper could perhaps be changed to include the editing aspect, since this is a definite contribution of the approach, and somewhat compensates for the computation costs of building the multiplane representations.

Experiments show the method works well compared to other approaches, and the ablations effectively show the influence of the various parts of the method.

**Weaknesses:**

An obvious cost of the proposed approach is that it requires depth information as input (e.g. an RGBD image). The paper states "The input depth map here can be easily estimated using off-the-shelf depth estimators, and the actual cost is comparable to pre-instance mask generation in existing mainstream mask-guided instance matting methods." No support is given for these claims in the paper, however. How do errors in the depth map affect performance? (i.e. how sensitive is the approach to depth map errors?)
No information is given in the experiment section how the value of N (number of depth planes) was set. There should be an ablation or hyperparameter search to determine an appropriate value.

The paper states "i ranging from 1 to S representing instance-level plane for each distinct human instances within the image", but does not say where these human instances come from. Are these detected by some external semantic (to identify humans) instance segmentation algorithms? If so, which ones were used? In the experiment section all that was mentioned is that certain datasets were used, and my assumption is that the ground truth instance labels were used as input. But some guidelines should be given on how the instances (and depth maps) would be generated when presented with unseen data which were not in the training datasets.
Also, no information or experiments are given as to the effect of errors in the instance segmentation on the performance of the approach.

In Table 1, both headings are labeled "instance agnostic". One of the these (the lower one) should be instance dependant.

A minor point, but the proposed method is not limited to human matting, so it is unclear why the authors chose to focus the paper on human matting. Nothing in the approach is specific to human instances. Any other type of instance (e.g. horses) could just as well be used, and would give the method more generality and applicability. There needs to be specific information which is unique to human instances for the approach to really be called human instance matting. Otherwise, it is just instance matting.
As an example of this point consider the paper by Tan et al (WACV 2018 - which should be referenced as an early work in this area) states "To simplify the task, we restrict our problem by focusing on human instance composition, because human segments exhibit strong correlations with their background and because of the availability of large annotated data. " This motivates the approach based on the focus on human instances, which the current paper under review does not do at all.
Tan et al. "Where and who? automatic semantic-aware person composition." 2018 IEEE Winter Conference on Applications of Computer Vision (WACV). IEEE, 2018.
The conclusion does shed some light on this, stating "our work mainly focuses on human instances. This is partially due to the lack of data for multi-instance matting for other categories.", but this does not mean that the approach is necessarily a human instance matting technique. It could be used for non-human matting if such training data was used.

In section 2.1 the paper writes "Previous image matting (...) is conducted under the single-instance assumption without multi-instance awareness...". This is not true. There are previous works that do multi-instance matting.
For example, the introduction of the paper writes: "InstMatte (Sun et al., 2022) first introduced the fomulation of multi-instance matting...". But even this statement is not true. Sun et al 2022 was not the first paper to formulate multi-instance matting. For example, the paper:
Hu and Clark. "Instance segmentation based semantic matting for compositing applications." 2019 16th Conference on Computer and Robot Vision (CRV). IEEE, 2019 uses instance segmentation to delineate foreground objects to be matted, and uses the simple iterative approach of matting these one at a time, with the other instances temporarily being considered as background.

There should be references to the early papers on semantic matting, as the more recent ones build on the ideas from these. Besides the two papers mentioned earlier, the following paper should be mentioned:
Sun,  Tang, and Tai. "Semantic image matting." Proceedings of the IEEE/CVF Conference on Computer Vision and Pattern Recognition. 2021.

There are two references for the same paper in the (Qinglin Liu, Shengping Zhang, Quanling Meng, Bineng Zhong, Peiqiang Liu, and Hongxun Yao. End-to-end human instance matting. IEEE Transactions on Circuits and Systems for Video Technology, 2023.) One should be deleted.

**Questions:**

See the weaknesses section for questions to be answered in the author responses.

---

### Official Review · Reviewer_dZDg · 2024-11-04

**Soundness:** 3
**Presentation:** 3
**Contribution:** 3
**Rating:** 6
**Confidence:** 3

**Summary:**

The authors presented a MPI combined with instance matting method to represent image, and shows its generalization capability for instance-aware image editing. The paper is mainly focusing on human-background representation. The architecture is a transformer-based network which consists of plan generation network and refinement module. Strong supervision makes it generate both the color and instance masks all together, to enable the model to do zero-shot image editing easily. The idea is intuitive and interesting.

**Strengths:**

- The idea of combining MPI with instance matting is intuitive and effective.
- The zero-shot performance on instance editing task is convincing.

**Weaknesses:**

- The paper is more about human matting and editing, so better to indicate that in the title to avoid confusion and overclaiming.
- The generation quality of occluded areas, including occluded people or complicated background, can be not good enough, since the model is a transformer-based framework, but not trained in a diffusion-based way.
- The model relies on the off-the-shelf depth estimator to obtain accurate depth information as 3D geometry. So its performance is highly relaying on the depth map accuracy. It is somehow a constraints which may limit the model performance.

**Questions:**

- Does the training dataset contain de-occluded instance mask and RGB contents? Why the model can learn the de-occluded contents without special supervision? Is it because the dataset is synthetic?
- Is it possible to show each layer in the MPI after learning? It will be interesting to visualize each layer in images.
- How can the model be generalized to object matting in longer term? Is it possible to extend it to a diffusion-based framework for better generation quality in the future?

---

### Meta-Review · Area_Chair_QZzf · 2024-12-22

**Metareview:**

The paper presents MP-Mat, a framework using dual multiplane representations for human instance matting, improving occlusion handling and boundary disentanglement. It achieves state-of-the-art results in matting and excels in zero-shot image editing, outperforming specialized methods. The approach is efficient, versatile, and demonstrated through extensive experiments.

Strengths:
Novel Approach: Combines depth-based and instance-level multiplane representations, effectively handling occlusions and complex boundaries.
Strong Results: Achieves state-of-the-art performance in matting and surpasses specialized methods in zero-shot image editing.
Innovative Features: Introduces background as a special instance and uncertainty-guided refinement for improved efficiency and clarity.

Weaknesses:
Depth Dependency: Heavy reliance on external depth estimators, with limited analysis of error sensitivity and plane number impact.
Limited Scope: Validation is restricted to human instances, lacking experiments on other object categories or datasets.
Incomplete Evaluation: Missing key metrics (e.g., Grad, Conn) and challenging visual examples to fully assess matting quality.
Lacking References: Overlooks prior works in multi-instance matting and makes overstated claims without sufficient justification.

Despite some weaknesses, all reviewers give positive scores based on the paper's contributions.
We are glad to accept this paper.

**Additional Comments On Reviewer Discussion:**

The reviewers raised two primary concerns: (1) the model's robustness across different depth estimators (raised by dZDg, dGBZ, ibAt, Yuvj) and (2) its generalization ability to non-human instances (raised by dZDg, dGBZ, Yuvj). The authors provided thorough responses, which satisfied these concerns.

Additionally, specific reviewer feedback was addressed:

Reviewer ibAt was convinced by the clarifications and raised their score to Accept.
Reviewer Vmaq also expressed satisfaction and increased their score from negative to positive.
The detailed rebuttals and improved scores indicate that the authors successfully addressed the key issues, supporting the decision to accept.

---

### Decision · Program_Chairs · 2025-01-22

Accept (Poster)